# Relative and Chronological Age in Successful Athletes at the World Taekwondo Championships (1997–2019): A Focus on the Behaviour of Multiple Medallists

**DOI:** 10.3390/ijerph19031425

**Published:** 2022-01-27

**Authors:** Gennaro Apollaro, Yarisel Quiñones Rodríguez, Tomás Herrera-Valenzuela, Antonio Hernández-Mendo, Coral Falcó

**Affiliations:** 1School of Sport Sciences and Exercise, Faculty of Medicine and Surgery, University of Rome Tor Vergata, 00133 Rome, Italy; gen.2012.ita@hotmail.com; 2Department of Sports Dididactics, University of Pinar del Río Hermanos Saíz Montes de Oca, Pinar del Río 20100, Cuba; yariselqr@gmail.com; 3Sciences of Physical Activity, Sports and Health School, Faculty of Medical Sciences, Universidad de Santiago de Chile (USACH), Santiago 9170022, Chile; tomas.herrera@usach.cl; 4Department of Social Psychology, Social Anthropology, Social Work and Social Services, University of Málaga, 29071 Málaga, Spain; mendo@uma.es; 5Department of Sport, Food and Natural Sciences, Western Norway University of Applied Sciences, 5020 Bergen, Norway

**Keywords:** long-term development, weight category, career, talent, longevity

## Abstract

The aims of this study were to investigate the relative and chronological age among taekwondo world medal winners (by gender, Olympic 4-year period, Olympic weight category; *N* = 740), and to study the behaviour of multiple medallists (*N* = 156) to monitor changes in weight categories and wins over time. The observed birth quartile distribution for the heavyweight category was significantly skewed (*p* = 0.01). Female athletes (22.2 ± 3.5 years) achieve success at a significantly younger age (*p* = 0.01) than their male counterparts (23.6 ± 3.3 years). In the weight categories, female flyweights were significantly younger than those welterweights (*p* = 0.03) and heavyweight (*p* = 0.01); female featherweights were significantly younger than those heavyweights (*p* = 0.03). Male flyweights and featherweights were significantly younger than those welterweights and heavyweights (*p* = 0.01). When a taekwondo athlete won a medal several times, he/she did so within the same Olympic weight category group and won two medals in his/her career (*p* = 0.01). Multiple medallists of the lighter and heavier groups did not differ in the number of medals won but in the time span in which they won medals (*p* = 0.02). The resources deployed by stakeholders to achieve success in these competitions highlight an extremely competitive environment. In this sense, the information provided by this study can be relevant and translated into key elements.

## 1. Introduction

The current popularity of taekwondo is well demonstrated by the more than 200 national member associations of World Taekwondo (WT) and its confirmation as one of the core sports in the upcoming Paris 2024 Olympic Games (OG) [1,2]. Firstly, these aspects are of considerable importance if we consider that the foundation of the WT dates back less than 50 years (28 May 1973) and the first edition of the World Championships (WC) coincides with this historical date (25–27 May 1973) [1,3]. Secondly, taekwondo contested its first official Olympic Games in 2000 at the Sydney Olympics [4]. The reconfirmations over the past 20 years as an Olympic sport are the result of the WC’s efforts to make significant changes to the rules that have made taekwondo more dynamic [5,6], and the transition from using a manual to an electronic scoring system that has made it a more equal sport [7].

The differences between the WC and the OG of taekwondo are not limited to the different time paths. The WC has been held every two years since 1973, but the competition for females was introduced 14 years later in the eighth WC in 1987 [3]. From this edition and still today, eight weight categories per gender are contested in each World Championship [8]. The 2019 WC saw the participation of 975 athletes from 150 countries [3], so the athletes participating in the WC generally represent the different nations equally. The electronic body device first appeared in the WC in the 2009 edition and the electronic head device in the 2015 edition [3]. In contrast, the OG is held every four years with two editions of the WC held within each 4-year period. Since its first Olympic edition in Sydney 2000, taekwondo has been contested for both males and females in four weight categories per gender [8], as there is a halving of the number of weight categories compared to the WC due to a grouping of them. Although the qualification system for the Taekwondo OG has undergone changes over time, such as the use of world ranking after the London 2012 OG [8], athletes generally represent continents equally. The electronic body protector and the electronic helmet protector were introduced at the London 2012 and Rio 2016 OG, respectively [4].

The WC and the OG can be considered the most important indicators of success for national taekwondo federations, as the long-term development programmes put in place are explicitly targeted at these competitions [9,10]. In this direction, the increasing interest in taekwondo on the part of scientific research [11,12] has so far provided an important body of knowledge that has translated into key elements for those stakeholders (e.g., coaches, physical trainers, sports managers) who operate on the front line of development programmes at the highest level [13,14,15]. In this context, it must be pointed out that research in taekwondo on extremely topical issues, such as the influence of athletes’ growth and maturation on selection and development processes at the highest level, as well as the relationship between age and the most important sporting competitions is relatively lacking [16,17,18] when compared to other sporting contexts [19,20]. Both areas provide valuable information regarding the individual development of the athlete as well as the social-cultural influence on his/her formation [21].

The first area leads to the phenomenon known as the relative age effect (RAE). RAE indicates an over-representation of athletes born at the beginning of the selection year within youth sports competitions, using the common practice of grouping athletes into bi(annual) age categories [19,22,23]. This system implies that an athlete may be relatively older or younger than his or her peers, and research has shown that these differences are associated with immediate and long-term consequences [24,25]. Specifically, it is hypothesised that relatively older athletes have greater cognitive and physical maturity than their relatively younger peers [19], resulting in greater opportunities for selection into developmental programmes, access to competition and a greater likelihood of becoming high-level athletes in the long-term [26,27]. Taekwondo, in its youth competitions in the cadet and junior categories, also uses a grouping system involving possible differences in age up to 24 months with the selection year starting on 1 January and ending on 31 December of the following year [8]. In this context, no previous study has analysed the RAE in the cadet and junior categories of taekwondo. On the contrary, Albuquerque et al. [16] and Campideli et al. [17] provided the first overview of this phenomenon in taekwondo by analysing athletes who took part in Olympic competitions also to understand what happens or has happened in previous age categories. However, the analysis of the RAE in these studies did not involve weight categories.

Regarding the age of peak competitive performance, it can be said that it is specific to the sporting context, as it is influenced by the skills and attributes required for success in a particular event [20]. In some sports (e.g., track and field, running, swimming) the age of peak performance is identified in correspondence with the athletes’ best performances and records [20,28,29]. In other sports such as soccer, Oterhals et al. [30] recently proposed the use of a new proxy based on age at the nomination for major individual awards to determine the average age at peak individual soccer performance. In combat sports, since performance cannot be quantified in seconds or centimetres, the WC and the OG can be considered the two most important indicators of peak performance [31]. Apollaro and Ruscello [18] found that female and male Olympic taekwondo athletes were successful at an average age of 23.3 ± 3.2 and 24.4 ± 3.3 years, respectively. At the same time, they found no age difference in competitive achievement (i.e., between medallists, losers in repechage and bronze medal matches; losers in the preliminary rounds), but when examining weight categories, they found that generally lighter athletes were younger than heavier ones.

The above framework leads to two important aspects. Firstly, the evident lack of studies in taekwondo on these highly topical issues and how they have only affected athletes who have participated in the OG. Secondly, the need to fill these gaps as the current popularity of taekwondo and the resources deployed by stakeholders outlines an extremely competitive environment at the highest level and the consequent need to clarify these aspects. Therefore, the main objective of this study is to explore the relative and chronological age of successful athletes at the World Taekwondo Championships, from 1997 to the 2019 edition, to provide important information about this competition that will celebrate 50 years of history with the 2023 edition. Specifically, (1) to investigate the relative age effect among world medal winners, by overall sample, gender, Olympic 4-year period and Olympic weight category; (2) to examine the chronological age at which taekwondo athletes win medals at the World Championships, to compare it by gender, Olympic 4-year period and Olympic weight category, to check whether there was a change in the age category distribution over the Olympic 4-year periods; (3) to study multiple world medallists to monitor changes in weight categories and wins over time.

## 2. Materials and Methods

### 2.1. Participants and Data Collection

The study included 740 medallists (females = 373; males = 367) of the 12 World Taekwondo Championships between 1997 and 2019 (from the overall of 768 medallists, representing 96% of the total population). Birth dates and weight categories were collected from publicly available online sources (https://www.taekwondodata.com/resultlist_select.html (accessed on 6 June 2021)). The use of data from open-access sites has been previously described in other studies [18,31] and there are no ethical issues involved in the analysis and interpretation of the data used as these were obtained in a secondary form and not from direct experimentation.

### 2.2. Procedures

Regarding the analysis of the RAE, the birth month of each world medal winner was used to define the birth quartiles (BQ). Several athletes won a medal in more than one edition of the WC and, in some cases, competed in different weight categories. The first medal won by the athlete in the WC was selected to avoid repetitive data. The calendar year (from 1 January to 31 December) was divided into four BQ: BQ1 = January, February and March, BQ2 = April, May and June, BQ3 = July, August and September, BQ4 = October, November and December. Analyses were carried out by an overall sample of gender (female and male), Olympic 4-year period (Sydney 2000, Athens 2004, Beijing 2008, London 2012, Rio 2016 and Tokyo 2020. The details of the WC grouping are shown in Table 1A) and Olympic weight category (flyweight, featherweight, welterweight and heavyweight. The details of the World weight categories grouping are shown in Table 1B).

Regarding the analysis of the chronological age, considering that each edition of the WC may take place at a different time of year, the age of each world medallist was calculated as follows: year of the WC edition-year of birth of the world medallist = age of world medallist. World medallists were divided by gender (female and male), Olympic 4-year period (Sydney 2000, Athens 2004, Beijing 2008, London 2012, Rio 2016 and Tokyo 2020; Table 1A), Olympic weight category (flyweight, featherweight, welterweight and heavyweight; Table 1B) and age group (<20 years, 20–25 years, 25–30 years and >30 years).

### 2.3. Statistical Analysis

Data were tabulated and organised in a Microsoft Excel worksheet and then reported and analysed using IBM SPSS Statistics for Windows, version 26.0 (IBM Corp., Armonk, NY, USA). Regarding the analysis of the RAE, chi-square (χ^2^) goodness-of-fit tests were performed to compare the observed BQ distribution against the expected BQ distribution. Similar to other studies [16,25,32], the sample consisted of international athletes so the expected values were calculated based on the assumption of a uniform distribution of births in each quartile of the year because the distribution of births is influenced by environmental and cultural factors [33]. Effect size (ES) was reported using Cramer’s *V* as small (0.06–0.17), medium (0.18–0.29) and large (>0.30) [34]. When the initial χ^2^ goodness-of-fit test was found to be statistically significant, odds ratios (OR) and 95% confidence intervals (95% CI) were calculated for the quartiles (BQ1, BQ2 and BQ3) with the youngest group used as reference (BQ4) [19,25]. Regarding the analysis of the chronological age, for each variable, the normality of the data distribution was examined using the Shapiro-Wilk test and Q-Q plots. Descriptive statistics (mean ± standard deviation [95% CI]) were used to summarise the collected data. A Student’s *t*-test for independent samples was used to compare female and male world medallists. ES was reported using Cohen’s *d* as trivial (0.0–0.19), small (0.2–0.59), moderate (0.6–1.1), large (1.2–1.9) and very large (>2.0) [35]. As this preliminary analysis revealed age differences between genders, each gender was analysed separately [18,31]. Thus, for each gender, a two-way ANOVA (Olympic 4-year period and Olympic weight category) was performed with Tukey’s posthoc tests, when appropriate, to detect significant differences in the age of world medallists from different groups. ES was reported using partial eta squared (η^2^_p_) as trivial (η^2^_p_ < 0.01), small (0.01 < η^2^_p_ < 0.06), moderate (0.06 < η^2^_p_ < 0.14) and large (η^2^_p_ > 0.14) [36]. Where χ^2^ tests of association were performed to identify the association between Olympic 4-year periods and the age groups. Where χ^2^ goodness-of-fit tests and OR (when appropriate) with 95% CI were performed to study the multiple world medallists for consequent analyses. For these analyses, ES was reported using Cramer’s *V*. Statistical significance was accepted at *p* < 0.05.

## 3. Results

The observed BQ distribution for the heavyweight category was significantly skewed (χ^2^_(3)_ = 10.81; *p* = 0.01; *V* = 0.17, small) when compared to the expected BQ distribution. The ORs show how podiums of WC are over-represented by medal winners from BQ3 in the heavyweight category (BQ1 vs. BQ4: OR = 1.35, 95% CI = 0.63–2.88; BQ2 vs. BQ4: OR = 1.90, 95% CI = 0.92–3.94; BQ3 vs. BQ4: OR = 2.20, 95% CI = 1.07–4.52). On the contrary, medal winners from BQ4 are under-represented on podiums of WC in the heavyweight category. The complete results from the χ^2^ goodness-of-fit-tests are shown in Table 2.

Female world medallists were younger (22.2 ± 3.5 [95% CI = 21.8–22.6] years) than male world medallists (23.6 ± 3.3 [95% CI = 23.3–23.9] years) (t_738_ = 5.83; *p* = 0.01; *d* = 0.40, small).

For female world medallists, main effects were found by Olympic 4-year period (F_5,349_ = 2.45; *p* = 0.03; η^2^_p_ = 0.03, small) and Olympic weight category (F_3,349_ = 6.57; *p* = 0.01; η^2^_p_ = 0.05, small; Table 3A). Female world medallists in the Sydney 2000 4-year period were younger (*p* = 0.03) than those in the Tokyo 2020 4-year period. Female world medallists in the flyweight category were younger than those in the welterweight (*p* = 0.03) and heavyweight (*p* = 0.01) categories. Female world medallists in the featherweight category were younger than those in the heavyweight (*p* = 0.03) categories (Table 3A).

For male world medallists, there was no main effect by Olympic 4-year period (F_5,343_ = 2.07; *p* = 0.07; η^2^_p_ = 0.03, small), but main effects were found by Olympic weight category (F_3,343_ = 14.25; *p* = 0.01; η^2^_p_ = 0.11, moderate; Table 3B). Male world medallists in the flyweight category were younger than those in the welterweight (*p* = 0.01) and heavyweight (*p* = 0.01) categories. Male world medallists in the featherweight category were younger than those in the welterweight (*p* = 0.01) and heavyweight (*p* = 0.01) categories (Table 3B).

Considering the main effects found for the weight categories, we then considered all the multiple world medallists (those who won two or more medals in WC) for further analysis [18]. This study included 156 medal winners for a total of 379 medals out of 740 (51%).

For the first analysis, we compared multiple world medallists who kept the same Olympic weight category over time with multiple world medallists who changed the Olympic weight category over time. We also compared multiple world medallists who won two medals with multiple world medallists who won three or more medals. A difference was observed between multiple world medallists who kept the same Olympic weight category over time and multiple world medallists who changed Olympic weight category over time (85% vs. 15%; χ^2^_(1)_ = 77.56; *p* = 0.01; *V* = 0.71, large; OR = 5.78, 95% CI = 3.36–9.95). At the same time, a difference was observed between multiple world medallists who won two medals and multiple world medallists who won three or more medals (71% vs. 29%; χ^2^_(1)_ = 27.92; *p* = 0.01; *V* = 0.42, large; OR = 2.47, 95% CI = 1.55–3.94).

In the second analysis, the multiple world medallists were divided into two groups on the basis of the main effects found for the weight categories (“lighter”: flyweight and featherweight, “heavier”: welterweight and heavyweight) and the multiple world medallists who changed category over time were placed in a group taking into consideration their last win. Considering the multiple world medallists who won two medals and the time span in which they won them, we also compared the lighter group with the heavier group. Considering the multiple world medallists who won three or more and the time span in which they won them, we compared the lighter group with the heavier group. The results of the second analysis are shown in Figure 1.

No association was found between Olympic 4-year period and age group for both females (χ^2^_(15)_ = 15.22; *p* = 0.44; *V* = 0.12, small) and males (χ^2^_(15)_ = 12.60; *p* = 0.63; *V* = 0.11, small) (Figure 2).

## 4. Discussion

The main objective of the present study was to explore relative and chronological age in successful athletes at the World Taekwondo Championships from 1997 to 2019. This is the first time that relative and chronological age have been analysed in athletes from the world’s top competition. Analyses were carried out by gender, Olympic 4-year period and Olympic weight category. In addition, multiple world medallists were taken into account to monitor changes in weight categories and wins over time.

RAE was not observed among world medal winners in the overall population, by gender and by Olympic 4-year period. Albuquerque et al. [16] explored RAE in Olympic Taekwondo athletes from Sydney 2000 to Beijing 2008. In line with our results, RAE did not emerge among these athletes in the overall population by gender and by single editions. Subsequently, Campideli et al. [17] analysed RAE in taekwondo considering the London 2012 and Rio 2016 OG. A significantly skewed distribution emerged only for males in Rio 2016, with a higher percentage of athletes in BQ3. One of the main hypotheses discussed in previous research to justify the absence of RAE, in sports and between genders, concerns the concept of competitiveness [16,25,32]. Competitiveness is defined by the number of available athletes and the popularity of the sport in a given country and, according to Musch and Grondin [23], could be a necessary and important condition for RAE to be present in sports contexts. Taekwondo entered the Olympic circuit relatively later than most Olympic combat sports [37] and also started contesting WC later than combat sports such as wrestling, judo and karate [3]. In this context, Albuquerque et al. [16] hypothesised how an increasing number of countries and athletes will increase competitiveness in taekwondo over time, as well as the potential of RAE when evaluated over the long-term. The result observed in male taekwondo athletes in Rio 2016 would seem to provide the first evidence to support this hypothesis. However, according to Campideli et al. [17], this result must be interpreted with caution as some countries adopt July as the start of the selection year [19,23]. At the same time, the impact of the RAE should be greater in countries that are traditionally more competitive in specific Olympic sports (e.g., Korea in taekwondo, Japan in judo, Cuba in boxing) [25]. Recently, Minsoo et al. [38] compared the BQ distribution of 12,054 Taekwondo athletes registered with the Korean Sports Association with that of a sample of the general Korean population. A statistically significant difference emerged in the distribution of BQ that confirmed the presence of RAE, with an over-representation of taekwondo athletes in BQ1 compared to BQ4.

The main results were in the weight category, where an over-representation of world medal winners emerged in BQ3 in the Olympic heavyweight category. Firstly, Musch and Grondin [23] and Cobley et al. [19] argued that grouping athletes into weight categories could attenuate the presence of RAE in combat sports because many of the physical advantages of more mature athletes would be controlled through this practice. Secondly, the importance of stratifying RAE research by weight category in combat sports was emphasised, as the different physical-physiological demands between weight categories and between sports (striking and grappling combat sports [14]) might facilitate the presence of RAE in some weight categories over others [32,39,40]. Albuquerque et al. [39], studying RAE in Olympic judo weight categories, found an over-representation in BQ3 compared to BQ4 in male athletes in the half-heavyweight category and a classical view of RAE in male athletes in the heavyweight category. Albuquerque et al. [32] extended this analysis to a larger sample of Olympic judo and confirmed the presence of RAE in male athletes in the heavyweight category. These authors hypothesised that in weight categories where there are no upper limits, the influence of the RAE is probably greater because athletes mature earlier and may be more likely to excel in crucial periods of sporting development, which would translate into a selection of athletes who were born in the first semester. The results of a meta-analysis on RAE in combat sports suggest that it is unclear how weight categories can eliminate RAE and highlight that this phenomenon does not occur consistently [25].

In this context, our study is the first to have investigated the presence of RAE in taekwondo weight categories, moreover in a population of medallists only. Firstly, the competitiveness of the sport [23] and the hypothesis of outcome uncertainty [41] could also justify our weight category results in taekwondo world medal winners. Generally, while the athletes taking part in the OG and WC represent different nations and continents equally, the same may not be true for medallists [42,43,44]. Consequently, the possible over-representation on the podiums of some nations could influence the presence of RAE in medallist populations. Chaplin and Mendoza [45] analysed competitive balance in boxing at the Commonwealth Games and provided evidence of a steady deterioration of competitive balance since the 1990s. Zheng et al. [43], analysing the competitive balance in table tennis in the OG and WC (1988–2016), identified China as dominant, and successful female athletes as more dominant than their male counterparts. In this context, Kazemi et al. [46] showed that Korea was most successful in taekwondo at the Sydney 2000 OG and the presence of RAE was confirmed in Korean taekwondo [38]. Therefore, an analysis of competitive balance in taekwondo could provide important information about competitiveness and success in this sport, and help to understand the results of the present study. Secondly, the unclear understanding of RAE in combat sports [25] and the weak correlation between RAE and maturity status in young athletes [47,48] support the importance of relating RAE, maturity status and physical performance. Recently, Radnor et al. [49] found that maturity status and relative age were differently associated with sprint performance in young soccer players. Particularly, advanced maturity was associated with superior sprint performance in most age groups, whereas relative age was, in most of the cases, unrelated to sprint performance. Thus, the authors defined RAE and maturity status as two distinct constructs, supporting the hypothesis that relative age does not necessarily imply more advanced maturity [49,50].

Regarding chronological age, female taekwondo athletes (22.2 ± 3.5 years) succeeded at the WC at a significantly younger age than their male counterparts (23.6 ± 3.3 years). Franchini et al. [31], analysing the age at which judo athletes participated in WC and OG (1993–2018), reported that female athletes (24.9 ± 3.9 years) were significantly younger than their male counterparts (25.4 ± 3.8 years). Apollaro and Ruscello [18], exploring the age at which taekwondo athletes competed in the OG (2000–2016), found a significant difference between the ages of female athletes (23.8 ± 4.1 years) and male athletes (25.1 ± 3.9 years). It is important to observe that Olympic female and male taekwondo athletes were successful at an average age of 23.3 ± 3.2 and 24.4 ± 3.3 years, respectively [18]. The fact that taekwondo athletes generally seem to achieve success at a younger age in WC than in the OG is in agreement with the results in judo. In fact, Franchini et al. [31] also highlighted that judo athletes were generally older in the OG than in most WC and that younger athletes seemed to compete in WC in the first year of Olympic preparation. The authors hypothesised how high-level judo athletes seem to direct their careers to achieve the highest competitive level at the OG. Therefore, the younger age of successful taekwondo athletes at the WC compared to the OG provides insights into the importance of the OG also within this combat sport. For this purpose, future studies should extend these analyses to the entire population of athletes who took part in the World Taekwondo Championships.

Within the Olympic 4-year periods, female world medallists in Sydney 2000 were significantly younger than those in Tokyo 2020. Taking into account the same time span included in our study, in the World Judo Championships, the female athletes who participated in the 2001 and 2013 editions were significantly younger than the athletes who took part in the 2015 edition [31]. At the Olympic level, in both taekwondo and judo, no significant differences were found for female athletes between the Sydney 2000 and Rio 2016 OG [18,31]. With regard to male world medallists, no significant differences in age have emerged within the Olympic 4-year periods. In contrast, age differences over time were found for male judo athletes who participated in the 1997 edition, as they were significantly younger than those who took part in the 2007, 2011 and 2015 editions. The athletes who took part in the 2001 edition were significantly younger than the athletes who took part in the 2011 edition, while the athletes who took part in the 2010 edition were significantly younger than those in the 2011 and 2015 editions [31]. Additionally, in males, no significant difference emerged at the Olympic level, both in taekwondo and judo [18,31]. Firstly, these results show that significant increases in age over time have generally influenced WC over the last six Olympic 4-year periods and in the different populations analysed. Secondly, the gradual increase in the age of female world medallists over time (culminating in the significant difference between Sydney 2000 and Tokyo 2020) is in line with the increases found by Barreira et al. [21] in women’s football. The authors hypothesised that the sports development system is providing better opportunities and support for female players by enabling them to prolong their careers. Moreover, by studying the age at peak performance of several Olympic events over a fairly wide time span, Elmenshawy et al. [51] highlighted the importance of the changes in societal expectations and climates toward women in the past 40–50 years. According to the authors, those social changes may have allowed female athletes to continue competing at older ages, contrary to earlier years where female athletes typically would have quit competing upon marriage and/or childbirth [51]. Thirdly, the age of female world medallists was for the first time close to that of males in the Tokyo 2020 Olympic 4-year period. Elmenshawy et al. [51] reported a similar tendency for the age of peak performance in the modern era for male and female medallists. According to the authors, there might be a biological peak performance age that is applicable to both males and females.

The main results were in the weight category where, regardless of gender, lighter world medallists are generally younger than heavier ones. Female world medallists in the flyweight category were younger than those in the welterweight and heavyweight categories. Female world medallists in the featherweight category were younger than those in the heavyweight category. Male world medallists in the flyweight and featherweight categories were younger than those in the welterweight and heavyweight categories. Our results are in line with the main effects found in the weight categories in judo, considering both the WC and the OG [31]. For example, the female judo athletes in the extra-lightweight category were younger than those from the lightweight to the heavyweight category. Male judo athletes in the half-lightweight category were younger than those from the half-middleweight to the heavyweight category. Male judo athletes in the half-heavyweight and heavyweight categories were older than those from the lightweight to the middleweight category. One of the hypotheses, formulated by Franchini et al. [31] to justify their data, is that high-level judo athletes change their weight category as they age, rather than continuing to engage in rapid weight loss procedures or to accommodate the potential for age-related decline in physical abilities. However, the authors pointed out the absence of publications showing this change in weight categories or the effects of this change on performance at the highest level of the sport. Subsequently, Apollaro and Ruscello [18] also identified similar effects in Olympic taekwondo weight categories. Female athletes in the flyweight category were younger than those in the heavyweight category. Male athletes in the flyweight and featherweight categories were younger than those in the welterweight and heavyweight categories.

To this purpose, Apollaro and Ruscello [18] carried out an additional analysis that included all taekwondo athletes who participated in two or more editions of the OG to monitor changes in weight categories over time. This analysis showed that generally when a taekwondo athlete reaches Olympic competition more than once in his career, he competes in the same weight category and participates in two consecutive editions. Athletes in the lighter categories seem to have more difficulty reaching Olympic competition more than two consecutive times in their career. In contrast, two non-consecutive participations and three or more consecutive participations seem to be more characteristic of athletes in the heavier categories [18]. According to Apollaro and Ruscello [18], these results may provide more than an explanation for the fact that lighter athletes are generally younger than heavier athletes in the Olympic taekwondo population, rather than the common observation that athletes change their weight category as they age. In line with this analysis, and considering the main effects found in the weight categories, another objective of our study was to monitor all the world’s multiple medallists, who reflect 51% of the medals awarded between 1997 and 2019. The most important results show that, generally, when a taekwondo athlete wins a medal several times at the WC, he or she does so within the same Olympic weight category group and wins two medals in his or her career. Secondly, multiple medallists of the lighter and heavier groups do not differ in the number of medals won but in the time span in which they win medals. Specifically, multiple medallists of the lighter group seem to find it more difficult to win two or three medals in a time span longer than two Olympic 4-year periods. On the contrary, winning two or three medals over two or more Olympic 4-year periods seems to be more characteristic of multiple medallists of the heaviest group. This twofold and divergent aspect found in our results seems to be in line with the behaviour of multiple Olympians of different weight groups and provides insights into the younger age found in successful lighter athletes in the World Taekwondo Championships compared to heavier athletes.

Apollaro and Ruscello [18] justified their results by hypothesising that Olympic athletes when they fail to maintain the Olympic weight category of their first participation, compete in adjacent World weight categories during the 4-year period, and then return to the same Olympic weight category in the next Olympic edition and maintain a likely competitive advantage [8]. Consequently, the possible difficulties resulting from such long-term weight management and the increased competitiveness in terms of participants in the selection process could represent a barrier for lighter athletes but translate into longer Olympic careers for heavier athletes [18]. Secondly, on the basis of the differences in technical-tactical behaviour at the London 2012 Olympic Games between athletes in the lighter categories and athletes in the heavier categories [52], Apollaro and Ruscello [18] highlighted the likely long-term physiological impact that these aspects could exert on the duration of athletes’ careers. In the World competition, Bridge et al. [53] showed that the activity profile of the fight is modulated by the athlete’s weight category. During the semifinals and finals of the 2005 WC, athletes in the heavyweight category performed fighting activities for longer periods than athletes in the finweight category due to more prohibited acts during each exchange. This seemed to involve a consequent increase in the number of general stoppages after each exchange. In contrast, athletes in the featherweight category sustained preparatory activity for longer periods than the finweight and heavyweight categories with fewer interruptions from exchanges and non-preparatory actions, resulting in less recovery from the demands of both activities [53]. Santos et al. [54], studying the semifinals and finals of the 2007 WC, identified an increase in the number of attacks and a reduction in balancing time during the third round in all weight categories. The reduction in balancing time affected the male finweight, flyweight and bantamweight weight categories and the female welterweight, middleweight and heavyweight categories. One possible explanation, according to Santos et al. [54], is that the average weight between these two groups of categories was similar, suggesting that the decline in balancing time may be weight dependent.

Lastly, world taekwondo medallists showed no significant changes in age groups in both genders. World medallists are most represented in the 20–25 age group with very similar percentages between genders. In female world medallists, the <20 age group has a double value of the 25–30 age group, whereas in male world medallists the <20 age group is lower than the 25–30 age group. In the World Judo Championships, a significant decrease in the percentage of female athletes in the <20 age group was identified during the 2015, 2017 and 2018 editions, preceded by changes in the 25–30 age group. For male judo athletes, there has been a significant increase in the >30 age group since the 2001 edition, in parallel with changes in the 20–25 age group. In Olympic judo, only for female athletes, there has been a significant increase in the 25–30 age group and a significant decrease in the 20–25 age group over time [31]. In Olympic taekwondo, in line with our results, there was no significant change in the different age groups in both genders. In this population, athletes are most represented in the 20–25 age group with the same percentage in both genders [18]. The absence of significant differences in the distribution of age groups in the Olympic taekwondo population was justified by the authors by the fact that taekwondo entered the Olympic circuit relatively later than most Olympic combat sports [37] and that the absence of changes in the different age groups will have to be interpreted in the long-term, taking into consideration future taekwondo participations in the OG [18]. Despite the WC also started to be contested later than combat sports [3], the absence of differences in the distribution of age groups over time (in parallel with the only difference found over time in the average age within the Olympic 4-year periods) must be interpreted with caution as it could represent a specific characteristic of this population. However, the present study is the only one to have carried out a detailed age analysis only on medallists in a combat sport. This encourages to focus not only on those participating in WC and OG but also specifically on successful athletes in an attempt to delineate their characteristics and differences.

### 4.1. Limitations and Future Research Lines

Although this study was the first to analyse relative and chronological age in athletes from the world’s top competition and to investigate the presence of RAE in taekwondo weight categories, some limitations should be considered. We have only considered WC medallists for analysis as birth dates, which can be retrieved from publicly available online sources, are limited to this population. As a result, WC editions and World weight categories have been grouped by Olympic 4-year periods and Olympic weight categories, respectively. The possibility in the future of involving the entire population of WC participants in the analyses would allow (1) a clearer view by edition and by World weight category; (2) to study the temporal interaction between the WC and the OG, as well as the careers progression of high-level taekwondo athletes. However, a detailed analysis of only successful athletes highlighted (1) that not using country-specific expected date-of-birth distributions in international samples (adjustment used, as births are influenced by environmental and cultural factors) [33] could have an even greater impact on medallist populations, as the competitive balance is sport-specific [43,44]; (2) the caution in interpreting the results as they could represent a specific characteristic of this population. The differences in the time span of victories of multiple medallists of different weight groups could partially justify the fact that, generally, lighter world medallists are younger than heavier ones. To this purpose, other factors could contribute to these differences, such as (1) the long-term impact of the rapid weight loss process and the use of aggressive methods, especially at the beginning of one’s career [31,55,56]; (2) the uniqueness of the context analysed, as the weight management during the Olympic 4-year period can be influenced by the specific regulations of each discipline [8,18]. Secondly, this different distribution of successes over time deserves further investigation, as factors such as long-term weight management, competitiveness and the impact of technical-tactical behaviour are only the first hypotheses, but other factors (or a combination of these) could play a key role.

### 4.2. Pratical Applications

The results of our study, when combined, provide insights for stakeholders involved in selection and development processes at the highest level. For example, the influence of growth and maturation of athletes could be investigated by analysing the relationship between biological maturation (rather than relative age) and sport-specific physical performance [47,48,49,50]. Maturity status should be considered when comparing physical performance scores to ensure stakeholders are not comparing early and late maturers within the same age group [49]. Secondly, the important age differences found in the weight categories cannot be addressed only by individualising training programmes. The negative consequences of the factors discussed behind these differences (in particular, long-term weight management and competitiveness) presuppose a solid and supportive talent development environment for the athletes [57] in order to avoid phenomena, such as early dropping out.

## 5. Conclusions

Female athletes achieve success at the World Taekwondo Championships at a younger age than their male counterparts. An increasing age trend has emerged in the female world medallists over the last two decades. The age differences between the lightest and heaviest medallists, in combination with the differences between multiple medallists in these groups in the time span of their wins, add insights into longevity at the highest level. The study of the relative age effect in this population highlights the need to continue to investigate this phenomenon in taekwondo to clarify the influence of growth and maturation on selection and development processes at the highest level. Taekwondo will be one of the core sports in the next Olympic Games in Paris 2024 and the World Taekwondo Championship will celebrate 50 years of history with the 2023 edition. The resources deployed by stakeholders to achieve success in these competitions highlight an extremely competitive environment. In this sense, the information provided by this study can be relevant and translated into key elements.

## Figures and Tables

**Figure 1 ijerph-19-01425-f001:**
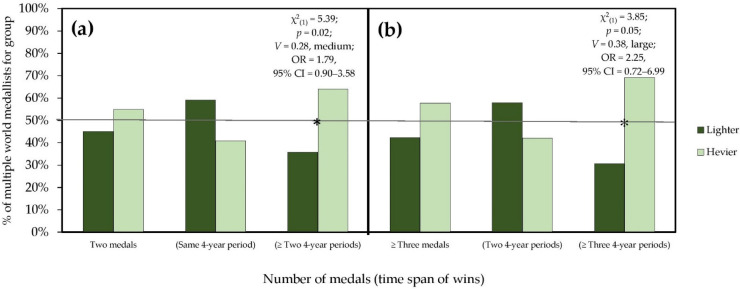
Evaluation of the weight groups of the world’s multiple medallists (*N* = 156) by the number of medals won and the time span of these wins. (**a**) No difference was observed between the lightest and heaviest multiple world medallists in winning two medals (χ^2^_(1)_ = 1.09; *p* = 0.30; *V* = 0.10, small). Regarding the time span, no difference was observed in winning two medals in the same 4-year period (χ^2^_(1)_ = 1.46; *p* = 0.23; *V* = 0.18, small). On the contrary, a significant difference was observed in winning two medals over two or more 4-year periods. (**b**) No difference was observed between the lightest and heaviest multiple world medallists in winning three or more medals (χ^2^_(1)_ = 1.09; *p* = 0.30; *V* = 0.16, small). Regarding the time span, no difference was observed in winning three or more medals in two 4-year periods (χ^2^_(1)_ = 0.47; *p* = 0.49; *V* = 0.16, small). On the contrary, a significant difference was observed in winning three or more medals over three or more 4-year periods. The grey line at 50% indicates the expected percentage of multiple world medallists under the null hypothesis that the weight category group has no effect on the number of medals won and on the time span of these wins. * Significantly skewed when compared to the expected distribution. Same 4-year period: 2 years; ≥two 4-year period: 2 to 14 years; two 4-year period: 2 to 6 years; ≥three 4-year period: 8 to 14 years.

**Figure 2 ijerph-19-01425-f002:**
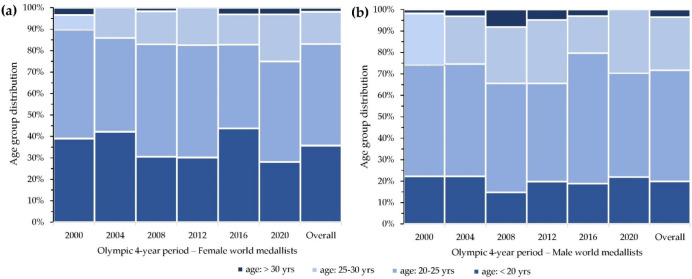
Age group distributions over time for (**a**) female and (**b**) male world medallists by Olympic 4-year periods (2000–2020).

**Table 1 ijerph-19-01425-t001:** (**A**) The details of the World Championships (WT) editions grouped by Olympic 4-year periods and (**B**) the details of the World weight categories grouped by Olympic weight categories.

**(A)** **Olympic 4-Year Period**			
**Sydney 2000**	**Athens 2004**		**Beijing 2008**
1997 Hong Kong WC (19–23 November)	2001 Jeju WC (1–7 November)		2005 Madrid WC (13–17 April)
1999 Edmonton WC (2–6 June)	2003 Garmisch-Partenkirchen WC (24–28 September)		2007 Beijing WC (18–22 May)
**London 2012**	**Rio 2016**		**Tokyo 2020**
2009 Copenhagen WC (14–18 October)	2013 Puebla WC (15–21 July)		2017 Muju WC (24–30 June)
2011 Gyeongju WC (1–6 May)	2015 Chelyabinsk WC (12–18 May)		2019 Manchester WC (15–19 May)
**(B)** **Olympic Weight Category**			
**Flyweight**		**Featherweight**	
Finweight		Bantamweight	
(−50 kg M/−43 kg F, until the 1997 edition)		(54.1–58 kg M/47.1–51 kg F, until the 1997 edition)	
(−54 kg M/−47 kg F, until the 2007 edition)		(58.1–62 kg M/51.1–55 kg F, until the 2007 edition)	
(−54 kg M/−46 kg F, until the 2019 edition)		(58.1–63 kg M/49.1–53 kg F, until the 2019 edition)	
Flyweight		Featherweight	
(50.1–54 kg M/43.1–47 kg F, until the 1997 edition)		(58.1–64 kg M/51.1–55 kg F, until the 1997 edition)	
(54.1–58 kg M/47.1–51 kg F, until the 2007 edition)		(62.1–67 kg M/55.1–59 kg F, until the 2007 edition)	
(54.1–58 kg M/46.1–49 kg F, until the 2019 edition)		(63.1–68 kg M/53.1–57 kg F, until the 2019 edition)	
**Welterweight**		**Heavyweight**	
Lightweight		Middleweight	
(64.1–70 kg M/55.1–60 kg F, until the 1997 edition)		(76.1–83 kg M/65.1–70 kg F, until the 1997 edition)	
(67.1–72 kg M/59.1–63 kg F, until the 2007 edition)		(78.1–84 kg M/67.1–72 kg F, until the 2007 edition)	
(68.1–74 kg M/57.1–62 kg F, until the 2019 edition)		(80.1–87 kg M/67.1–73 kg F, until the 2019 edition)	
Welterweight		Heavyweight	
(70.1–76 kg M/60.1–65 kg F, until the 1997 edition)		(+83.1 kg M/+70.1 kg F, until the 1997 edition)	
(72.1–78 kg M/63.1–67 kg F, until the 2007 edition)		(+84.1 kg M/+72.1 kg F, until the 2007 edition)	
(74.1–80 kg M/62.1–67 kg F, until the 2019 edition)		(+87.1 kg M/+73.1 kg F, until the 2019 edition)	

M: Male; F: Female; WC: World Championships.

**Table 2 ijerph-19-01425-t002:** Evaluation of quartiles of world medal winners’ birth by overall sample, gender, Olympic 4-year period and Olympic weight category.

	**Number and (%) of World Medal Winners for Birth Quartile**					**χ^2^_(df)_**	** *p* **	**Cramer’s *V***
	**BQ1**	**BQ2**	**BQ3**	**BQ4**	**Total**			
**Overall**	127 (25)	126 (24)	143 (28)	121 (23)	517	2.11_(3)_	0.55	0.04
**By gender**								
Female	67 (26)	61 (23)	71 (28)	61 (23)	260	1.11_(3)_	0.76	0.04
Male	60 (23)	65 (25)	72 (29)	60 (23)	257	1.51_(3)_	0.68	0.04
**By Olympic 4-year period**								
Sydney 2000	17 (16)	33 (32)	29 (28)	25 (24)	104	5.39_(3)_	0.15	0.13
Athens 2004	26 (29)	22 (25)	25 (28)	16 (18)	89	2.73_(3)_	0.44	0.10
Beijing 2008	23 (30)	18 (24)	14 (18)	21 (28)	76	2.42_(3)_	0.49	0.10
London 2012	21 (27)	12 (16)	20 (26)	24 (31)	77	4.10_(3)_	0.25	0.13
Rio 2016	20 (20)	21 (21)	36 (38)	21 (21)	98	7.22_(3)_	0.07	0.16
Tokyo 2020	20 (27)	20 (27)	19 (26)	14 (20)	73	1.36_(3)_	0.72	0.08
**By Olympic weight category**								
Flyweight	36 (26)	34 (25)	29 (21)	38 (28)	137	1.31_(3)_	0.73	0.06
Featherweight	38 (28)	25 (18)	36 (27)	36 (27)	135	3.10_(3)_	0.38	0.09
Welterweight	26 (22)	29 (25)	34 (30)	27 (23)	116	1.31_(3)_	0.73	0.06
Heavyweight	27 (21)	38 (29)	44 (34)	20 (16)	129	10.81_(3)_	0.01 *	0.17

* Significantly skewed when compared to the expected distribution; BQ: birth quartiles.

**Table 3 ijerph-19-01425-t003:** Descriptive statistics (m ± sd [95% CI]) of the chronological age (years) of (**A**) female and (**B**) male world medallists according to Olympic 4-year period and Olympic weight category.

**(A)** **Female World Medallists (*N* = 373)**					
**Olympic 4-Year Period**					
Sydney 2000 (*N* = 59)	Athens 2004 (*N* = 64)	Beijing 2008 (*N* = 59)	London 2012 (*N* = 63)	Rio 2016 (*N* = 64)	Tokyo 2020 (*N* = 64)
21.3 ± 3.7 a	21.6 ± 3.2	22.5 ± 3.4	22.5 ± 3.4	22 ± 3.5	23.1 ± 3.5
[20.4–22.2]	[20.8–22.4]	[21.6–23.4]	[21.7–23.3]	[21.1–22.9]	[22.2–24]
**Olympic weight category**					
Flyweight (*N* = 94)	Featherweight (*N* = 93)		Welterweight (*N* = 94)	Heavyweight (*N* = 92)	
21.1 ± 3.6 b	21.8 ± 3.1 c	22.5 ± 3.4		23.2 ± 3.5	
[20.4–21.8]	[21.2–22.4]	[21.8–23.2]		[22.5–23.9]	
**(B)** **Male World Medallists (*N* = 367)**					
**Olympic 4-year Period**					
Sydney 2000 (*N* = 54)	Athens 2004 (*N* = 63)	Beijing 2008 (*N* = 61)	London 2012 (*N* = 61)	Rio 2016 (*N* = 64)	Tokyo 2020 (*N* = 64)
23.4 ± 3.2	23.1 ± 3.2	24.5 ± 3.5	24.1 ± 3.5	23.2 ± 3.1	23.5 ± 3.2
[22.6–24.3]	[22.3–23.9]	[23.6–25.4]	[23.2–25]	[22.4–24]	[22.7–24.3]
**Olympic weight category**					
Flyweight (*N* = 91)	Featherweight (*N* = 94)		Welterweight (*N* = 88)	Heavyweight (*N* = 94)	
22.5 ± 3.2 d	22.8 ± 2.9 e		24.2 ± 3.2	25 ± 3.2	
[21.8–23.2]	[22.2–23.4]		[23.5–24.9]	[24.4–25.7]	

(a) significantly younger when compared to Tokyo 2020 (*p* = 0.03) 4-year period; (b) significantly younger when compared to welterweight (*p* = 0.03) and heavyweight (*p* = 0.01) categories; (c) significantly younger when compared to heavyweight (*p* = 0.03) category; (d) significantly younger when compared to welterweight (*p* = 0.01) and heavyweight (*p* = 0.01) categories; (e) significantly younger when compared to welterweight (*p* = 0.01) and heavyweight (*p* = 0.01) categories.

## Data Availability

Not applicable.

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
