# Peer review of "Relative and Chronological Age in Successful Athletes at the World Taekwondo Championships (1997–2019): A Focus on the Behaviour of Multiple Medallists"

_ijerph, 2022, doi:10.3390/ijerph19031425_

Round 1
Reviewer 1 Report
To the authors,
This was a very well written paper, with thorough understanding of previous work in this area, and good approach in analysis and discussion.
I support the publication of this work, but highly suggest adding a section about relevance to the athlete development staff. Going into detail about how this information can be applied to athlete development is critical in my opinion. This could be achieved by offering strategies toward athlete selection, training development, preparation toward competitions etc.
Thanks for you work, and I look forward to reviewing this minor modification.
Reviewer 2 Report
why don't you reflect about the importance of doing this kind of studies through maturation methods, in order to solve the stupidity of RAE hypothesis?
By the same logic, why don't you take the hypotheis of precocity of female maturation (relative to male) to discuss your between genders age results?
why don't you hipothize that results found in females about category weight and age may have the same problem as RAE hypothesis?
